# Restrictive versus liberal fluid resuscitation strategy, influence on blood loss and hemostatic parameters in mild obstetric hemorrhage: An open-label randomized controlled trial. (REFILL study)

Pim B. B. Schol[1,2]*, Natascha M. de Lange[1,2¤], Mallory D. Woiski[3], Josje Langenveld[2], Luc J. M. Smits[4], Martine M. Wassen[2], Yvonne M. Henskens[5], Hubertina C. J. Scheepers[1,6]

1 Department of Obstetrics and Gynecology, Maastricht University Medical Centre, Maastricht, The Netherlands, 2 Department of Obstetrics and Gynecology, Zuyderland, Sittard-Geleen, The Netherlands, 3 Department of Obstetrics and Gynecology, Radboud University Medical Centre, Nijmegen, The Netherlands, 4 Department of Epidemiology, Caphri School for Public Health and Primary Care, Maastricht, The Netherlands, 5 Central Diagnostics Laboratory, Maastricht University Medical Centre, Maastricht, The Netherlands, 6 GROW: School for Oncology and Developmental Biology and Department of Obstetrics and Gynecology, Maastricht University Medical Centre, Maastricht, The Netherlands

¤ Current address: Department of Obstetrics and Gynecology, Isala, Zwolle, The Netherlands
* pbbschol@hotmail.com

## Abstract

### Background

Evidence for optimal hemostatic resuscitation in postpartum hemorrhage (PPH) is lacking. Liberal fluid administration may result in acidosis, hypothermia and coagulopathy.

### Objective

We hypothesize that in early PPH a restrictive fluid administration results in less progression to moderate PPH.

### Study design

In four Dutch hospitals we recruited women of 18 years and over, and more than 24 weeks pregnant. Exclusion criteria were: anticoagulant therapy, known coagulation disorders, pre-eclampsia, antenatal diagnosis of abnormally adhesive placenta, and a contraindication for liberal fluid therapy. We blindly randomized participants at 500 mL and ongoing blood loss in the third stage of labor between restrictive fluid administration (clear fluids 0.75–1.0 times the volume of blood lost) and liberal fluid administration (clear fluids 1.5–2.0 times the volume of blood lost). The primary outcome was progression to more than 1000 mL blood loss. Analyses were according to the intention-to-treat principle.

**Data Availability Statement:** All relevant data are within the manuscript and its Supporting Information files.

**Funding:** The author(s) received no specific funding for this work.

**Competing interests:** The authors have declared that no competing interests exist.

## Results

From August 2014 till September 2019, 5190 women were informed of whom 1622 agreed to participate. A total of 252 women were randomized of which 130 were assigned to the restrictive group and 122 to the liberal group. In the restrictive management group 51 of the 130 patients (39.2%) progressed to more than 1000 mL blood loss versus 61 of the 119 patients (51.3%) in the liberal management group (difference, -12.0% [95%-CI -24.3% to 0.3%], p = 0.057). There was no difference in the need for blood transfusion, coagulation parameters, or in adverse events between the groups.

## Conclusions

Although a restrictive fluid resuscitation in women with mild PPH could not been proven to be superior, it does not increase the need for blood transfusion, alter coagulation parameters, or cause a rise in adverse events. It can be considered as an alternative treatment option to liberal fluid resuscitation.

## Trial registration

NTR3789.

## Introduction

Postpartum hemorrhage (PPH) is one of the most common reasons for peripartum intensive care unit (ICU) admittance and the main cause of maternal death worldwide. In high resource countries an increase in incidence of PPH is observed [1–4]. In the Netherlands the incidence of PPH rose from 4.1% in 2000 to 6.4% in 2013 [5].

Evidence for optimal hemostatic resuscitation in PPH is currently lacking [6]. The Managing Obstetric Emergencies and Trauma course (MOET) and the Royal College of Obstetricians and Gynecologists (RCOG) instructions advise generous volume resuscitation to restore blood volume and oxygen carrying capacity: about twice the lost volume and up to 3.5 L of fast fluid infusion in patients with more than 1000 mL blood loss or clinical shock [7, 8]. Dutch guidelines recommend to commence volume resuscitation at profuse blood loss, not disclosing a minimum amount of blood loss. This guideline states that there is currently no evidence to administer more fluids than lost [9].

Resuscitation with crystalloids and colloids have their own (dis)advantages and risks. Transfusing large amounts of crystalloids before commencing with blood products may result in acidosis, hypothermia and coagulopathy; the lethal triad [10]. Additionally, the hydroxyethyl starch solutions may impair clot function when used excessively [11].

Restrictive or permissive hypotension has been advocated as an alternative for liberal fluid resuscitation in other areas than obstetric care, although the amount of randomized controlled trials is limited. A restrictive fluid administration policy in other fields has shown to decrease further blood loss and the amount of blood transfused [12–17]. Physiological hemodynamic and hemostatic changes in pregnancy make that these result cannot be readily adopted in the postpartum hemorrhage care. During pregnancy plasma volume and red blood cell count increases 40% and 30% respectively. Cardiac output is increased and the blood pressure decreased in the second trimester and increased again at term [18, 19]. There is a hypercoagulable state during pregnancy which is most pronounced in the third trimester [20, 21]. Two

recent retrospective obstetric studies, showed that larger quantities of crystalloid volume administrated in the care of women with severe PPH was associated with a more severe deterioration of coagulation parameters. Fluid resuscitation with more than 4 L of crystalloid infusion was associated with more subsequent bleeding and adverse maternal outcomes (intensive care admittance, embolization, hysterectomy) [22, 23]. To date there are no publications on randomized trials on optimal fluid resuscitation in women with PPH.

The aim of this randomized controlled trial was to determine if a restrictive fluid administration policy in early and mild PPH (500 mL blood loss) leads to a decrease in progression to more than 1000ml blood loss compared to care as usual. We included both women with a caesarean and vaginal delivery. Although the risk of PPH is higher in women with a caesarean delivery, the question on optimal fluid administration is valid for both groups [24, 25]. We hypothesized a decrease in progression and therefore a decrease in adverse outcomes.

## Materials and methods

### Study design

REFILL was a randomized controlled multicenter trial performed from August 2014 until September 2019 in four hospitals in the Netherlands. This study was approved by the Medical Ethics Committee Maastricht University Hospital (NL4294206813). This trial is registered in the Netherlands Trial Register NTR3789 or NL3623 (date of registration, 11 January 2013). The study protocol was published in 2018 [12]. The study protocol is available online at: https://www.ncbi.nlm.nih.gov/pmc/articles/PMC5838856/

### Participants

Participants were recruited in four Dutch hospitals, two university hospitals (Maastricht University Medical Center, Radboud University Medical Center) and 2 regional teaching hospitals (Zuyderland Medical Center and Jeroen Bosch Hospital). Maastricht University Medical Center (MUMC) was coordinating center. All four centers have in-house midwives and medical residents (junior and senior), and a supervising gynecologist on call.

All women attending the outpatient clinic or the labor ward, and not in active labor, were considered for eligibility. These women were informed about the study if they met the inclusion criteria. The inclusion criteria were: age 18 years and over, understanding of the Dutch language, pregnant and labor starting after 24+0 weeks, and mentally competent. Both vaginal and caesarean deliveries were included. Exclusion criteria were: prophylactic or therapeutic anticoagulant therapy (carbasalate calcium within the previous 10 days or low molecular weight heparins within previous 48 h), known congenital coagulation disorders, pre-eclampsia, antenatal diagnosis of placenta accreta spectrum (due to likelihood of reaching primary outcome regardless of management), and contraindication for liberal fluid therapy (e.g. cardiac causes, systemic causes (Marfan), renal causes, pulmonary failure). We obtained written informed consent from all participants.

### Randomization and masking

Women who gave oral and written consent for the study were randomized if they reached 500 mL and ongoing blood loss postpartum. Enrolment was performed by the treating team of caregivers at that time through sealed opaque envelopes.

Treatment allocation was blinded by the use of sealed opaque envelopes including a trial number. Randomization was stratified per center, in blocks of four in an allocation of 1:1. Sequence was generated online (https://www.randomizer.org/) and the sealed opaque

envelopes were created by an independent research nurse or medical student not involved in the randomization of the patient. Local investigators were blinded to block size and allocation. Randomization envelopes were distributed per center by Maastricht University Medical Center.

## Procedures

The randomization envelopes were quickly and readily available on the labor ward or operating theatre. In case of 500 mL and ongoing blood loss at the third stage of labor an envelope was opened by the treating physician. The patient was either randomized to the restrictive fluid administration (intervention) group or to the liberal fluid administration (control) group. In the intervention group patients received fluids at 0.75–1.0 times the volume of blood loss. In the control group patients received 1.5–2.0 times the volume of blood loss. Blood loss was measured by weighing the absorption towels after childbirth. The first towel was disposed directly after childbirth and not measured as this also includes amniotic fluids. Blood loss during caesarean section was measured through suction and weighing operative gauzes after childbirth. The first 2000 mL of volume replacement consisted of NaCl 0.9% or Ringers lactate, or a combination of both on room temperature.

At 500 mL and ongoing blood loss allocation took place. Directly after randomization, moment T1 was initiated. If intravenous access was not yet present, intravenous access was established and blood samples for T1 were drawn. At T1 hemoglobin concentration, hematocrit, platelet count, activated partial thromboplastin time, prothrombin time, and fibrinogen concentrations were measured.

Hemodynamic parameters such as blood pressure and oxygen saturation were observed according to local protocol. Additional safety measures were taken in case of systolic blood pressure <90 mmHg, diastolic blood pressure <50 mmHg, a decrease of more than 20 mmHg in blood pressure, or a maternal heart rate of 125 beats per minute or more. In this case 500 mL additional volume was administered in 15 minutes in either group.

The second evaluation, T2, was 45–60 minutes after T1. At T2 a second set of blood samples as stated above were drawn. At T3, 12–18 hours post-partum a third set of blood sample was drawn for hemoglobin and hematocrit analysis only, if patients were still admitted to hospital.

If the patient reached 1500 mL blood loss the study protocol was exited and the patient was treated according to local massive obstetric hemorrhage protocol which also includes the blood transfusion policy. Blood samples as stated above were still drawn and patient data was analyzed on an intention-to-treat basis.

Third stage of labor was actively managed in all participants according to national protocol with the administration of 5 IE of oxytocin directly after childbirth and 10 IE oxytocin infused in 4 hours thereafter. Patients were kept warm. The underlying cause of the PPH was treated according to local and national guidelines (Dutch Society of Gynecology and Obstetrics, NVOG) [9].

All study parameters were collected from the patient chart and study files. The data were collected and stored anonymously in Maastricht University Medical Center in a restricted access file. A trial number was assigned to each patient at time of randomization.

## Outcomes

The primary outcome was the frequency of progression to major PPH (defined as blood loss > 1000 mL).

Secondary outcomes were the differences in hemoglobin concentration (mmol L$^{-1}$) 12–18 hours postpartum (including hemoglobin <5mmol L$^{-1}$), differences in transfusion requirements (number of units of packed red blood cells, fresh frozen plasma, thrombocytes, and

fibrinogen concentrate needed), differences in coagulopathies (platelets $<50.10^9 \, L^{-1}$, fibrinogen concentration $< 1g \, L^{-1}$ and activated partial thromboplastin time (APTT) and prothrombin time (PT) $> 1.5x$ mean control).

Severe adverse outcomes (SAE), defined as intensive care admittance, the need of four or more units of packed cells, embolization, and hysterectomy, were registered and analyzed by a data safety monitoring board (DSMB).

### Statistical analysis

Sample size was calculated with the assumption that, with standard care, around 30% of the women with 500 mL blood loss would progress to more than 1000 mL blood loss [12]. We calculated that, in order to be able to detect a 50% relative reduction, with a power of 0.80 and an alpha of 0.05, 118 patients would be needed in each study arm. To be able to compensate for incomplete data we aimed for 250 inclusions.

Comparative analysis was performed with either a Student's *t* test in case of continuous data or the chi-square test in case of dichotomous outcomes. Multivariable linear regression analysis was employed to check whether results were sensitive to controlling for baseline characteristics, including center of inclusion. Analyses were done according to the intention-to-treat principle. Missing data were scarce and not imputed. Analyses were performed using IBM SPSS 24.0 and SAS version 9.4.

A data safety monitoring board was established. The DSMB was notified at each SAE, after the first 2x 25 inclusions, and every 2x 50 inclusions thereafter for which they performed an interim analysis on the primary outcome and SAEs. Throughout the study there was no need to stop the trial prematurely.

The corresponding author had full access to all the data in the study and had final responsibility for the decision to submit for publication.

## Results

Between August 2014 and September 2019 5190 patients were assessed for eligibility of which 1622 patients gave informed consent to participate if they reached 500ml blood loss postpartum. A total of 252 patients were randomized, 130 were assigned a restrictive fluid administration strategy, and 122 a liberal fluid administration strategy (Fig 1). Maastricht University Medical Center recruited 74 participants, Radboud University Medical Center 37 participants, Zuyderland Medical Center 134 participants, and Jeroen Bosch Hospital recruited 7 patients.

Table 1 shows baseline characteristics, which were similar for the two groups. For all patients risk factors for PPH were evaluated (see S1 File for the risk factors collected). In two women no risk factors for PPH were present, the mean number of risk factors was 3 in both groups. In the liberal resuscitation strategy arm three patients discontinued treatment as they were diagnosed with pre-eclampsia. In the restrictive strategy arm no patient discontinued treatment. 130 patients in the restrictive fluid administration strategy, and 119 patients in the liberal fluid administration strategy were analyzed as intention-to-treat. The total mean crystalloid fluid administration after randomization in the restrictive arm was 1078 mL (SD1029 mL, median 800 (0–5200 mL)) and 1534 mL (SD 957 mL, median 1350 mL (0–4100 mL)) in the liberal arm (p = 0.000). All patients received crystalloids. Additional colloids were administered in 10/130 (7.7%, 48 mL SD 220) women in the restrictive arm versus 8/119 (6.7%, 47 mL SD 171) in the liberal arm (p = 0.957).

Table 2 shows both primary and secondary outcomes. In the restrictive policy 51 of the 130 patients (39.2%) progressed to more than 1000 mL blood loss versus 61 of the 119 patients (51.3%) in the liberal resuscitation policy arm (difference, -12.0% [95%-CI -24.3% to 0.3%],

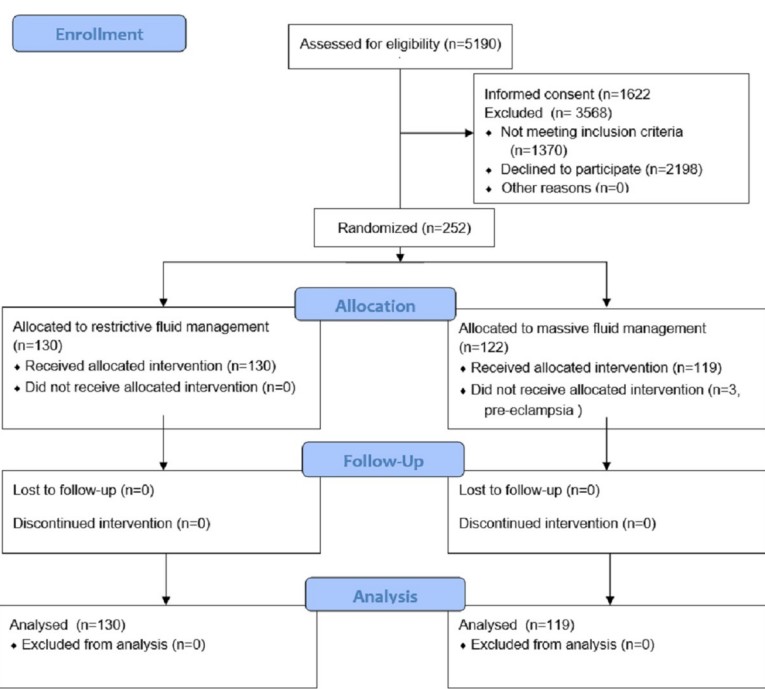

**Fig 1. Flow diagram.**

p = 0.057). Total blood loss did not differ significantly. Mean blood loss in the restrictive arm was 1182 mL (SD 761 mL) and in the liberal arm 1242mL (SD 621 mL), (p = 0.5).

There was no difference in hemoglobin (Hb) levels at T2, and T3 in the restrictive arm and liberal arm respectively. Hb levels $< 80.6$g L$^{-1}$ at T2 and T3 are comparable in the restrictive arm and liberal arm respectively.

The number of patients in need for blood products in both groups are comparable. Packed cells were primarily given to those exceeding the 1500 mL blood loss (n = 22/25), fresh frozen plasma and fibrinogen concentrate only in those exceeding 2000 mL of blood loss, and the thrombocytes were given in a case of 6000 mL blood loss. No significant difference in coagulopathy was observed between both policies: thrombocytes $<50.10^9$ L$^{-1}$, APTT and PT more than 1.5 times the reference range, and fibrinogen less than 1 g. The use of intrauterine balloon tamponade (n = 3/130 in the restrictive policy versus n = 2/119 in the liberal policy, p = 0.73) and the use of B lynch stitch (n = 2/130 in the restrictive policy versus n = 1/119 in the liberal policy, p = 0.61) are comparable. There was no use of arterial ligation in either group. Adverse events defined as ICU admittance, administration of more than 4 packed cells, embolization therapy, and hysterectomy were not different between both groups.

Causes identified for PPH are presented in Table 3. Main cause of PPH in both groups is uterine atony; 52.3% in the restrictive arm and 63.9% in the liberal arm.

Adjustment, by means of multiple regression, for the small differences in baseline characteristics (augmentation, episiotomy, analgesics) or controlling center of inclusion did not result in any meaningful changes in the effect estimates or in more precision.

## Discussion

### Principal findings

A restrictive fluid resuscitation in women with mild postpartum hemorrhage could not been proven to be superior (p = 0.057), even though the confidence interval around the effect estimation

**Table 1. Baseline characteristics.**

| | Restrictive (n = 130) | Liberal (n = 119) |
|---|---|---|
| Age (years) | 31.9 (±3.5; 22–41) | 31.6 (±4.5; 19–45) |
| BMI kg m$^{-2}$ | 25.6 (±5.5; 17–45) | 26·2 (±5.5; 17–47) |
| Gestational age (weeks) | 39.2 (±1·8; 30.4–42.3) | 39.3 (±1.6; 31.5–41·6) |
| Gravidity | 2 (1–7) | 2 (1–7) |
| Parity | 0 (0–4) | 1 (0–3) |
| Gestational age (days) | 276 (±12.7; 214–297) | 276 (±10.9; 222–293) |
| Risk factors HPP (amount) | 3 (0–10) | 3 (1–8) |
| **History of:** | | |
| Manual removal of placenta | 10 (7.7) | 4 (3.3) |
| Postpartum hemorrhage | 19 (14.6) | 15 (12.3) |
| Blood transfusion with postpartum hemorrhage | 8 (6.2) | 6 (4.9) |
| **Onset of labor** | | |
| Spontaneously | 31 (23.8) | 31 (25.4) |
| Induction | 83 (63.8) | 77 (63.6) |
| Caesarean, planned | 16 (12.3) | 13 (10.7) |
| **Pain relief** | | |
| Opioids | 24 (18.5) | 17 (13.9) |
| Epidural | 57 (43.8) | 63 (51.6) |
| No pain relief | 50 (38.5) | 28 (23.5) |
| **Outcome** | | |
| Delivery | | |
| Spontaneously | 89 (68.5) | 84 (68.9) |
| Ventouse | 13 (10) | 10 (8.2) |
| Caesarean | 28 (21.6) | 27 (22.1) |
| Caesarean, unplanned | 12 (9.2) | 19 (12.6) |
| Augmentation | 80 (61.5) | 84 (68.9) |
| Episiotomy | 46 (35.4) | 51 (41.8) |
| Vaginal rupture | 52 (40) | 48 (39.3) |
| Weight at birth (gram) | 3497 (±631; 1470–5130) | 3552 (±569; 1922–4740) |
| Macrosomia (>4kg) | 28 (21.5) | 26 (21.3) |
| **T1 laboratory parameters** | | |
| Hb (g l$^{-1}$) | 112.8 (±16.1; 61.2–143.4) | 112.8 (±12.9; 67.7–138.6) |
| Ht (%) | 0.33 (±0.05; 0.18–0.43) | 0.33 (±0.04; 0.21–0.42) |
| Thrombocytes (·10$^9$ l$^{-1}$) | 202 (±61; 75–379) | 198 (±48; 80–327) |
| APTT (sec) | 26.9 (±3.6; 21–48) | 26.6 (±3.7; 20.0–41.8) |
| Fibrinogen (g l$^{-1}$) | 4.2 (±0.9; 2–6.5) | 4.2 (±0.7; 2.2–5.9) |
| PT(sec) | 10.9 (±1.7; 9.1–16) | 10.8 (±1.8; 9–20) |

Data are n (%), mean (SD; range)

Gravity, parity and risk factors are presented as median (range)

BMI: body mass index, T1: time 1 at 500ml and start randomization, Hb: hemoglobin, Ht: hematocrit, APTT: activated partial thrombin time, PT: partial thrombin time

ranged from decreased progression risk by almost a quarter to near equality in the risk of progression (difference, -12.0% [95%-CI -24.3% to 0.3%]). A restrictive fluid resuscitation management in women with a moderate postpartum hemorrhage does not alter the need for blood transfusion, alter coagulation parameters, or cause a rise in adverse events. A more restrictive fluid resuscitation fluid management strategy could be a safe management choice in early and mild PPH.

**Table 2. Primary and secondary outcomes.**

| | Restrictive policy (n = 130) | Liberal policy (n = 119) | p |
|---|---|---|---|
| **Progression to more than 1000 mL blood loss** | 51 (39.2) | 61 (51.3) | 0.057 |
| **Total blood loss (mL)** | 1182 (761) | 1242 (621) | 0.5 |
| **Hemoglobin g l$^{-1}$** | | | |
| T2 | 105.5 (15.3) | 104.1 (15.6) | 0.652 |
| T3 | 92.7 (13.7) | 99.9 (83.8) | 0.849 |
| **Hemoglobin < 80.6g l$^{-1}$ (n)** | | | |
| T2 | 6 (4.6) | 7 (5.9) | 0.404 |
| T3 | 18 (13.8) | 18 (15.1) | 0.430 |
| **Transfusion (n)** | | | |
| Packed cells | 14 (10.8) | 11 (9.2) | 0.689 |
| Fresh Frozen Plasma | 3 (2.3) | 5 (4.2) | 0.397 |
| Thrombocytes | 1 (0.8) | 0 (0) | 0.338 |
| Fibrinogen | 2 (1.5) | 3 (2.5) | 0.581 |
| **Coagulopathy T2** | | | |
| Platelets <50.10$^9$ l$^{-1}$ | 0 | 0 | n/a |
| APTT > 1.5 times reference range (n (mean seconds)) | 3 (42.2) | 4 (43.0) | 0.858 |
| PT > 1.5 times reference range (n (mean seconds)) | 3 (14.6) | 4 (14.5) | 0.880 |
| Fibrinogen < 1 gram | 0 | 0 | n/a |
| **Adverse events (n)** | | | |
| ICU admittance | 1 | 1 | 0.157 |
| ≥ 4 packed cells | 1 | 2 | 0.223 |
| Embolization | 1 | 0 | 0.338 |
| Hysterectomy | 0 | 0 | n/a |

Data presented are n (%), mean (SD) unless otherwise stated

APTT: activated partial thrombin time PT: partial thrombin time ICU: intensive care unit T1: time 1 (at 500ml blood loss and start of randomization), T2: time 2 (45-60min after T1), T3: time 3 (12–18 hours after T1)

## Results

Outside the obstetric field there is still no widely implemented consensus on fluid management in peri-operative and trauma care. As outlined in our trial protocol there is little and contradictive evidence for either liberal or restrictive fluid resuscitation regimens [12]. In addition to this outline Myles et al reports an increased risk for acute kidney injury in high risk patients

**Table 3. Causes identified for PPH.**

| | Restrictive (n = 130) | Liberal (n = 119) |
|---|---|---|
| Uterine atony | 68 (52.3) | 76 (63.9) |
| Episiotomy | 32 (24.6) | 34 (28.6) |
| Retained placenta | 27 (20.8) | 26 (21.8) |
| Incomplete placenta | 8 (6.2) | 5 (4.2) |
| Cervical/vaginal trauma | 17 (13.1) | 11 (9.2) |
| Uterine rupture | 3 (2.3) | 0 |
| Inversio uteri | 0 | 0 |
| Coagulopathy | 0 | 0 |

Data are n (%)

during major abdominal surgery receiving a restrictive fluid management [26]. There was no difference in disability free survival in both groups. The randomized controlled trial of Myles et al. supports the dangers of hypoperfusion. However their study population is a high risk population undergoing major abdominal surgery which is not comparable to a relatively healthy obstetric population. Kwan et al reports, in their systematic review of six randomized controlled trials, no evidence for or against early or larger intravenous fluid administration strategies in uncontrolled hemorrhage in trauma patients [27]. No quantitative assessment could be provided due to diverse patient populations. They stress the necessity for further randomized controlled trials. We showed, in a systematic review, that a restrictive policy in elective surgery was favorable in comparison to a liberal fluid management policy for total complication rate, infection, and transfusion rate [13].

The lack of consensus on perioperative fluid management is reflected in the guidelines of the American Society of Anesthesiologists (ASA) and the European Society of Anaesthesiology (ESA). ASA has not updated their perioperative fluid management since 2008 [28]. In their perioperative blood management guidelines there is no mention of crystalloid or colloid use [29]. Their latest editorial note still marks the disagreement on the matter [30]. However they do agree that the optimal regimen is to replace the losses. The ESA state the lack of evidence to advice upon a perioperative fluid management. They do however advocate to avoid hypoperfusion, and advise a timely and aggressive stabilization of the cardiac preload taking a goal-directed approach [31]. However in their trauma guideline they recommend the use of a restrictive fluid management to achieve target blood pressure [32]. The ESA was also collaborator in the obstetric guideline of Network for the Advancement of Patient Blood Management, Hemostasis and Thrombosis (NATA) published in 2019. This multidisciplinary consensus statement recommend a restrictive fluid crystalloid administration of 1–2 mL crystalloids for every 1 mL blood lost [33]. This is a more liberal trend to how restrictive fluid management is generally advocated. Restrictive fluid resuscitation is based to replace the fluids lost with avoiding fluid overload [34].

The use of colloids can impair clot function by disturbing fibrin polymerization and by faster clot disintegration. Colloids may therefore increase blood loss [35–38]. The use of starches may increase the need for blood transfusion, increase the likelihood of acute kidney injury, and overall more side effects such as pruritis and rash [39–41].

Systematic reviews or randomized controlled trials evaluating fluid management protocols in the obstetric population are lacking. Our trend of a favorable outcome with a restrictive fluid management policy in the obstetric field is supported by the publications of Henriquez and Gillisen [22, 23]. In a retrospective cohort study Gillisen showed deterioration in coagulation parameters correlated with the amounts of crystalloid fluids infused. Levels of hemoglobin, hematocrit, platelet count, fibrinogen, and APTT were all negatively associated with the amount of crystalloid fluids infused. Henriquez performed a retrospective cohort study on women with a severe postpartum hemorrhage, finding that administration of more than 4 L of crystalloid fluids was independently associated with more maternal adverse outcomes (a composite of mortality and severe maternal morbidity defined as hysterectomy, embolization, or ICU admittance). Mean blood loss in both studies were 3.0 and 2.9 L respectively and exceeded our mean blood loss of 1.2 L.

## Clinical implications

Our study is prospective and data were gained in a randomized controlled setting. The results of this randomized controlled trial are applicable to a wide obstetric population as most women with PPH do not exceed the 1500ml of blood loss. These data can be used to design and perform new randomized controlled trials in this fragile population and acute setting.

## Strengths and limitations

To our knowledge this is the first randomized controlled trial on fluid resuscitation strategies within the obstetric field. We reached our calculated sample size in both resuscitation arms and had no loss to follow-up for the primary outcome. Baseline characteristics were well balanced and adjustment for small differences in the baseline gave similar results.

The randomization envelopes were available at a central point at the labor ward in each location. In case of a cesarian section, an envelope was taken to the operating theatre in case the patient would reach 500 mL blood loss. The operating theatres are not at the labor ward but in a different section of the hospital at all participating locations, making it impractical to pick up an envelope from the labor ward once the patient reached 500 mL blood loss. Unfortunately some of these unopened envelopes were disposed of during clean up instead of returned to their original central point, causing a slight imbalance of 130:122 in treatment assignments. This numerical imbalance does not influence validity. Even though we reached our aimed pre-calculated sample size, and the point estimate of the difference in risk of progression to more than 1000 mL blood loss pointed to a clinically relevant effect (absolute difference, 12.1%; number needed to treat, 9), the difference did not reach statistical significance (p = 0.057). With inclusion of a larger number of participants the power of study would have been higher. Arguably, in retrospect, for our sample size calculation we chose a minimally detectable relative risk that was too conservative (RR 0.5) with the consequence that smaller, but relevant, differences such as the one found would stay statistically non-significant. Pooling of our results (or data) with any similar future studies could yield more precision and enable smaller differences to be more easily detectable.

As this study was the first randomized controlled trial with a solely obstetric patient population, strict precautionary safety measures were in place. One of the main safety measure was abdication of the enrolled resuscitation arm when reaching 1500 mL blood loss. Therefore our results can only be applied to women with a PPH less than 1500 mL blood loss, which is the majority of all women experiencing PPH. Of all women in labor, 1.4–3.9% progress to more than 1500 mL blood loss [3, 42]. Another safety measure was the choice to commence resuscitation at 500 mL of blood loss. Signs of clinical shock can present as early as 750 mL of blood loss, defined as class II hemorrhagic shock [32]. However these limitations do reduce the ability to compare the more serious adverse outcomes such as transfusion need, embolization, IC admittance and hysterectomy between these groups.

Our progression to more than 1000 mL blood loss was 39.2% in the restrictive management group and 51.3% in the liberal management group which is higher than the general percentage of women with a PPH of more than 1000 mL: 3–5.5% [33, 42, 43]. We contribute this to a selected population, we only selected women who had more than 500ml and ongoing blood loss whereas the general percentage applies to the whole population of women giving birth. This is reflected in the percentage in the population who gave informed consent: 6.3% (102/1622) had more than 1000 mL blood loss. This is slightly higher than the reported incidences. Although important, optimal management and the safety of restrictive management or even permissive hypotension in massive PPH, concerning only a small minority of women, is not the scope of the current study. We chose to study the more common and therefore more relevant effect on the effects of fluid resuscitation in mild PPH. Adequately managing mild PPH can improve care for a large group of women and may prevent progression.

## Conclusions

Although a restrictive fluid resuscitation in women with mild PPH could not been proven to be superior it does not increase the need for blood transfusion, alter coagulation parameters,

or cause an increase in adverse events and therefore can be considered as an alternative treatment option. This study does not allow comments on safety on restrictive management in cases with massive PPH. More randomized controlled trials on fluid resuscitation should be conducted in patients with PPH to establish a evidence based recommendations on this matter.

## Supporting information

**S1 Checklist. CONSORT 2010 checklist of information to include when reporting a randomised trial**\*.
(DOC)

**S1 File. Risk factors postpartum hemorrhage.** BMI: body mass index PPH: postpartum hemorrhage.
(DOCX)

**S2 File. Dataset.**
(SAV)

## Acknowledgments

We would like to thank drs. N.M.A.A. Engelen, dr J.M. Middeldorp and dr. A. Kessels, for participation in the DSMB and performing the interim analyses during the trial. Further we would like to thank all participating staff and research nurses involved in the trial.

## Author Contributions

**Conceptualization:** Natascha M. de Lange, Luc J. M. Smits, Yvonne M. Henskens, Hubertina C. J. Scheepers.

**Data curation:** Pim B. B. Schol.

**Formal analysis:** Pim B. B. Schol, Luc J. M. Smits, Hubertina C. J. Scheepers.

**Investigation:** Pim B. B. Schol, Mallory D. Woiski, Josje Langenveld, Martine M. Wassen.

**Methodology:** Natascha M. de Lange, Luc J. M. Smits, Hubertina C. J. Scheepers.

**Project administration:** Pim B. B. Schol.

**Writing – original draft:** Pim B. B. Schol.

**Writing – review & editing:** Natascha M. de Lange, Mallory D. Woiski, Josje Langenveld, Luc J. M. Smits, Martine M. Wassen, Yvonne M. Henskens, Hubertina C. J. Scheepers.

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
