## [Decision Letter · Decision Letter 0]

16 Mar 2021

PONE-D-21-04160

Restrictive versus liberal fluid resuscitation strategy, influence on blood loss and hemostatic parameters in mild obstetric hemorrhage: an open-label randomized controlled trial. (REFILL study)

PLOS ONE

Dear Dr. Schol,

Thank you for submitting your manuscript to PLOS ONE. After careful consideration, we feel that it has merit but does not fully meet PLOS ONE’s publication criteria as it currently stands. Therefore, we invite you to submit a revised version of the manuscript that addresses the points raised during the review process.

This is the first RCT regarding restrictive versus liberal fluid resuscitation strategy in the management of PPH. The article is clear and very interesting, even if results were found to be negative. Some methodological comments (see reviewer comments) decrease the interest and some modifications should be done in the manuscript, especially in methods. The flow diagram should be checked because there are some errors in the number of patient (see reviewer comments). Some details should also be added in discussion about limitations of the study. To conclude, the authors reported an interesting, but negative RCT, and a lot of work should be done before resubmission.

We look forward to receiving your revised manuscript.

Kind regards,

Guillaume Ducarme, MD, MSc, PhD

Academic Editor

PLOS ONE

Journal Requirements:

2. Please amend your list of authors on the manuscript to ensure that each author is linked to an affiliation. Authors’ affiliations should reflect the institution where the work was done (if authors moved subsequently, you can also list the new affiliation stating “current affiliation:….” as necessary).

Reviewers' comments:

Reviewer's Responses to Questions

**Comments to the Author**

1. Is the manuscript technically sound, and do the data support the conclusions?

Reviewer #1: Yes

Reviewer #2: Yes

Reviewer #3: Yes

2. Has the statistical analysis been performed appropriately and rigorously? 

Reviewer #1: Yes

Reviewer #2: No

Reviewer #3: Yes

3. Have the authors made all data underlying the findings in their manuscript fully available?

Reviewer #1: Yes

Reviewer #2: Yes

Reviewer #3: Yes

4. Is the manuscript presented in an intelligible fashion and written in standard English?

Reviewer #1: Yes

Reviewer #2: Yes

Reviewer #3: Yes

5. Review Comments to the Author

Reviewer #1: Pim Schol et al. presented here a very interesting randomized clinical trial regarding restrictive versus liberal fluid resuscitation strategy in the management of PPH.

This is the first RCT in obstetrics field ++

In a general way, this article is clear, and very interesting, even if results were found to be negative.

Please find some comments that should be taken into account:

Specific comments:

Methods section:

L142: Were there some women treated with 1 to 1.5 times the volume of blood loss?

L.148: please explain more precisely T1, T2 etc… it is unclear.

Please state if women were administered heating blanket in the management of PPH.

L161: how many women exited from the study?

L172: I think that “moderate” should be changed to “severe” ?

L177: what was the policy regarding transfusion of blood product?

L182: Is there a policy regarding the use of intrauterine tamponade?

Results section:

L219: May the authors provide administered volume of colloids?

Table 1: how can the authors explain the rate of episiotomy?

I do not understand the percentage of episiotomy 51 (41·8 %) ? Explain.

Please explain the rate of macrosomia that appears high…?

Table 1: same remark as above: explain the overall of 23 pregnancies…

May the authors indicate BMI, parity in the table?

Discussion section:

L268-270: please moderate this sentence since the benefits do not appear in this study.

L290-295: in which situation? PPH? Surgery with high-risk of bleeding?

L298-300: it is strange since their “restrictive policy” corresponds to twice the volume of blood loss whereas in your study, the restrictive policy corresponded to a 0.75-1 times the volume of blood loss. Therefore, it is difficult to compare. The authors should discuss the definition of “restrictive policy”

L305: are there data regarding colloids use on coagulation parameters (vs cristaloids)?

L307: please provide example.

Please provide few sentences about the use of intrauterine ballon that is not documented in the manuscript.

Reviewer #2: This is an interesting, but ultimately negative clinical trial. I believe the extreme sample size assumption that the investigators wished to see a 50% reduction was overly conservative, and resulted in an underpowered study. There was a trending difference between the groups in the primary outcome, but the sample size was not large enough to get a significant p-value. This should be noted in the "weaknesses" section as one possible reason for the negative primary outcome.

I have numerous comments on how the manuscript and analysis could be improved:

1. The abstract Methods section should be less about inclusion and exclusion and more about the study design.

2. There are no statistical methods listed in your "Statistical Methods" section. You only discuss sample size and what the DMC was monitoring. Without specifying how you are analyzing, sample size computations and monitoring are irrelevant.

3. Since you did not say how you analyzed the study, it appears that you did not conduct a stratified test, which every statistician would require following a stratified (by center) analysis. Please correct this, or specify that you did this already.

4. The DMC does not "abandon the trial prematurely", they determine if there is sufficient evidence to stop a trial early for efficacy, futility, and/or safety.

5. Throughout the paper you refer to the groups with the phrase "there is no difference". As you know, determination of a statistical difference requires more than just observation of numbers. It is more appropriate to discuss that the data appear to be comparable with respect to the given metric.

6. The discussion section should (as noted above) indicate whether the assumptions in the sample size and power computations were realized in the study (e.g., effect size, LTF assumptions, etc.)

7. I have no idea how you would up with a 130:122 imbalance in treatment assignments given that you were using permuted blocks with a fixed size of 4. The maximum imbalance possible is 2 due to an unfilled block, or where there non-compensating incomplete blocks in multiple centers.

8. You do not discuss the clinical centers, where they were, what their staffing was, how many patients they each recruited, and how study procedures were standardized.

9. Fixed block sizes are frowned upon by statisticians--I presume the investigators were blinded to the block size to mitigate selection bias, and that should be stated if so.

Reviewer #3: Dear Author,

I reported my comments on the manuscript titled “Restrictive versus liberal fluid resuscitation strategy, influence on blood loss and hemostatic parameters in mild obstetric hemorrhage: an open-label randomized controlled trial. (REFILL study)”.

I think that the topic is interesting. References list appears up to date and appropriate. The manuscript is clear and easy to understand also for non-specialized reader.

Without doubts, the study have many limits, some of which punctually explained in the manuscript. In my opinion a great limit is the selection of population, there are many confounding factor, for example the choose to recruit women both after vaginal delivery and cesarean section or the recruitment of women with risk factors for PPH. Probably the best way is recruit women without risk factors after a spontaneous vaginal delivery. Anyway, this limit is partially overcame by the fact that the sampling method is the same in both groups and by the decision to analyze data according to the intention-to-treat principle.

Before the acceptance I suggest some minor revisions.

You (at line 143-146) explain that blood loss was measured by weighing the absorption towel after childbirth, but the first towel was disposed directly after childbirth and not measured as this also includes amniotic fluid. I suggest to explain how do you have solved the problem of amniotic fluid interference in case of cesarean section, in which the blood loss was measured through suction and weighing operative gauze.

In the methods, at line 164, I suggest to explain the actions performed in the active management of third stage of labor, for example the type and dosage and route of administration of uterotonic agent.

I suggest to check the flow diagram in the figure 1, because in my opinion could be some errors in the number of patient. Authors say that they have assessed for eligibility 5190 women. After that 1622 women signed informed consent and 3568 women were excluded. In the reasons of exclusion we have 1370 women that not met inclusion criteria and 1946 that declined to participate. In this way we have 252 lacking women (1370+1946=3316, 252 women less of 3568 women excluded). I think that this 252 women are not the women randomized, because the 252 women randomized were in the group of women that signed the consent. So I suggest to check the data, because if the 252 women randomized are added to the women that signed the consent we have not 1622 women but 1874 women. In this way the authors have to check also the data at line 339-340, because the rate 102/1874 is 5,4% in line with the litterature data.

In the discussion at line 270, remove "low risk PPH", because the women in the study are not at low risk PPH. I think that you refer about the early and mild PPH (500 mL blood loss).

After these minor revisions, I suggest to accept this manuscript because in my personal opinion the paper is interesting for at least 3 reasons: the originality (first study on this topic), the study design (prospective, blinded, randomized controlled multicenter clinical trial) and for the results (which certainly deserve to be then confirmed on a larger scale).

6. PLOS authors have the option to publish the peer review history of their article (what does this mean?). If published, this will include your full peer review and any attached files.

Reviewer #1: No

Reviewer #2: No

Reviewer #3: No

---

## [Author Response · Author response to Decision Letter 0]

21 Apr 2021

PLOS ONE 

Senior Editorial Operations Manager R. Yterdal

1265 Battery Street

Floor 2

San Francisco, CA 94111

United States of America 

Maastricht, 18th April 2021

Dear dr. Ducarme,

Thank you and the reviewers for the investment in our manuscript. We reviewed the comments and revised our manuscript “Restrictive versus liberal fluid resuscitation strategy, influence on blood loss and hemostatic parameters in mild obstetric hemorrhage: an open-label randomized controlled trial. (REFILL study)” accordingly. 

As requested we supply a marked-up copy of our manuscript as well as an unmarked version. Below you will find the points raised and our response to it per point. 

We hope you will accept our submission for “PLOS ONE”.

Kind regards,

Pim Schol, M.D. 

 

We look forward to receiving your revised manuscript.

Kind regards,

Guillaume Ducarme, MD, MSc, PhD

Academic Editor

PLOS ONE

Journal Requirements:

 Checked

2. Please amend your list of authors on the manuscript to ensure that each author is linked to an affiliation. Authors’ affiliations should reflect the institution where the work was done (if authors moved subsequently, you can also list the new affiliation stating “current affiliation:….” as necessary).

Adjusted

Adjusted

Reviewers' comments:

Reviewer's Responses to Questions

Comments to the Author

1. Is the manuscript technically sound, and do the data support the conclusions?

Reviewer #1: Yes

Reviewer #2: Yes

Reviewer #3: Yes

2. Has the statistical analysis been performed appropriately and rigorously?

Reviewer #1: Yes

Reviewer #2: No

Reviewer #3: Yes

3. Have the authors made all data underlying the findings in their manuscript fully available?

Reviewer #1: Yes

Reviewer #2: Yes

Reviewer #3: Yes

4. Is the manuscript presented in an intelligible fashion and written in standard English?

Reviewer #1: Yes

Reviewer #2: Yes

Reviewer #3: Yes

5. Review Comments to the Author

Reviewer #1: Pim Schol et al. presented here a very interesting randomized clinical trial regarding restrictive versus liberal fluid resuscitation strategy in the management of PPH.

This is the first RCT in obstetrics field ++

In a general way, this article is clear, and very interesting, even if results were found to be negative.

Please find some comments that should be taken into account:

Specific comments:

Methods section:

L142: Were there some women treated with 1 to 1.5 times the volume of blood loss?

Women were randomized between 0.75-1.0 times and 1.5-2.0 times, there was no separate group of 1-1.5 times. We chose these groups so that the treatment would differ enough to establish difference. Women were analyzed according to intention to treat. However we did analyze ‘as treated’ which yielded no different results. 

L.148: please explain more precisely T1, T2 etc… it is unclear.

Adjusted. 

Please state if women were administered heating blanket in the management of PPH.

It is part of national protocol to keep women warm at >1000 mL blood loss: https://www.nvog.nl/wp-content/uploads/2018/02/bijlage-checklist-bij-uitgangsvragen-rl-HPP-2015lhee.pdf .This can be also be achieved with prewarmed blankets as well, not only heating blankets. We referenced to this national protocol at line 171 and added the statements “Patients were kept warm”. 

L161: how many women exited from the study?

None exited while randomized. In the analysis 3 women with pre-eclampsia during labor were omitted from analysis as this was an exclusion criteria. It is stated in the “Results” section, line 223

L172: I think that “moderate” should be changed to “severe” ?

Depending on the defining source. RCOG defines “minor” and “major”, with major subdivided into “moderate” (1000-2000mL) and “severe” (>2000mL). WHO defines “severe” as >1000mL. We adjusted it to “major”, this will suit both definitions. 

L177: what was the policy regarding transfusion of blood product?

According to national guideline (https://www.nvog.nl/wp-content/uploads/2018/02/bijlage-checklist-bij-uitgangsvragen-rl-HPP-2015lhee.pdf), ordering blood products at >1000mL and ongoing blood loss. Administering at 2000mL or when 2L of crystalloids are given. If patients reached 1500 mL of blood loss the randomized arm was exited and patients were treated according to local massive hemorrhage protocol, as stated in line 163. Two centers have thromboelastometry available which makes a patient directed blood transfusion possible beyond the standardized protocol. 

Clarified at line 164

L182: Is there a policy regarding the use of intrauterine tamponade?

Mentioned in national guideline (https://www.nvog.nl/wp-content/uploads/2018/02/Hemorrhagia-postpartum-HPP-3.0-14-11-2013.pdf) . To be used at physicians discretion.

Results section:

L219: May the authors provide administered volume of colloids?

Added, as the mean was very low with a larger standard deviation we initially chose to only mention the amount of patients that received the colloids. We now added the mean and standard deviation to the text. 

Table 1: how can the authors explain the rate of episiotomy?

I do not understand the percentage of episiotomy 51 (41·8 %) ? Explain.

Please explain the rate of macrosomia that appears high…?

This study was conducted in the hospital which results in a selected population. In the Netherlands the ultimately low risk pregnancies are guided by the midwives at home or in a birth center without medical supervision. Women were informed at our outpatient clinic and gave informed consent before start of the induction or spontaneous labor with a medical indication (macrosomia, intra-uterine growth retardation, cesarian section indication etc). Resulting in the higher need for intervention. Also an episiotomy and macrosomia are a risk factor for postpartum hemorrhage. Resulting in women with an episiotomy and/or macrosomia to be more easily randomized. 

Table 1: same remark as above: explain the overall of 23 pregnancies…

Overall 23 pregnancies? We do not understand this remark. The average number of pregnancies is 2. 

May the authors indicate BMI, parity in the table?

BMI and parity are already stated in the table at the top section. 

Discussion section:

L268-270: please moderate this sentence since the benefits do not appear in this study.

Moderated. 

L290-295: in which situation? PPH? Surgery with high-risk of bleeding?

Perioperative fluid management outside the obstetric field. Also indicated at start of the section. Clarified at these lines. 

L298-300: it is strange since their “restrictive policy” corresponds to twice the volume of blood loss whereas in your study, the restrictive policy corresponded to a 0.75-1 times the volume of blood loss. Therefore, it is difficult to compare. The authors should discuss the definition of “restrictive policy”

The overall basis of restrictive fluid policy is to replace the volume lost, to avoid fluid overload and hypoperfusion. It is indeed strange for the NATA to therefore recommend a more liberal approach to the restrictive policy. This underlines the point made, that there is no real consensus on fluid resuscitation policy. We added a reference and definition to lines 319-320

L305: are there data regarding colloids use on coagulation parameters (vs cristaloids)?

There are more than enough data on colloids disturbing the coagulation parameters, that is the reason crystalloids are preferred over colloids. We feel that such a detailed outline is beyond the scope of this article. We did however insert a short comment on the effect of colloids on coagulation. Hydroxyethylstarches (HES) are also prone to give more (acute) kidney injury and adverse events. 

L307: please provide example.

This is stated at this line prior to revisions: “In a retrospective cohort study Gillisen showed deterioration in coagulation parameters correlated with the amounts of crystalloid fluids infused. Levels of hemoglobin, hematocrit, platelet count, fibrinogen, and APTT were all negatively associated with the amount of crystalloid fluids infused”

We feel that elaborating with exact numbers on a research that is not our own, does not contribute to the readability of our discussion. We feel that we referenced correctly and when exact numbers are wanted, one should read the study itself to interpret the numbers correctly as this study provides a lot of data. 

Please provide few sentences about the use of intrauterine ballon that is not documented in the manuscript.

The use of intrauterine balloon tamponade is not one of the primary or secondary outcome measures. We added a sentence about this intervention on request. We also added the use of B lynch stich and arterial ligation to complete the interventions used. 

Reviewer #2: This is an interesting, but ultimately negative clinical trial. I believe the extreme sample size assumption that the investigators wished to see a 50% reduction was overly conservative, and resulted in an underpowered study. There was a trending difference between the groups in the primary outcome, but the sample size was not large enough to get a significant p-value. This should be noted in the "weaknesses" section as one possible reason for the negative primary outcome.

Added:

“Even though we reached our aimed pre-calculated sample size, and the point estimate of the difference in risk of progression to more than 1000 mL blood loss pointed to a clinically relevant effect (absolute difference, 12.1%; number needed to treat, 9), the difference did not reach statistical significance (p=0.057). With inclusion of a larger number of participants the power of study would have been higher. Arguably, in retrospect, for our sample size calculation we chose a minimally detectable relative risk that was too conservative (RR 0.5) with the consequence that smaller, but relevant, differences such as the one found would stay statistically non-significant. Pooling of our results (or data) with any similar future studies could yield more precision and enable smaller differences to be more easily detectable.” 

I have numerous comments on how the manuscript and analysis could be improved:

1. The abstract Methods section should be less about inclusion and exclusion and more about the study design.

We feel that a clear description of a performed study is crucial for further research. The inclusion and exclusion criteria are provided in lines 115 to 122. These are mentioned and not explained in this paragraph or thereafter in the methods section. The rest of the methods section is focused on the execution of the trial to allow for reproduction by others. 

2. There are no statistical methods listed in your "Statistical Methods" section. You only discuss sample size and what the DMC was monitoring. Without specifying how you are analyzing, sample size computations and monitoring are irrelevant.

We already mentioned the method for comparative analysis in our original manuscript (Lines 194-197) 

“Comparative analysis was performed with either a Student’s t test in case of continuous data or the chi-square test for dichotomous outcomes. Analyses were done according to the intention-to-treat principle. Missing data were not imputed. All analyses were performed using IBM SPSS 24.0 software.”

We added information on the additional sensitivity analysis that we carried out by means of multiple linear regression: ”Multivariable linear regression analysis was employed to check whether results were sensitive to controlling for baseline characteristics, including center of inclusion.” 

3. Since you did not say how you analyzed the study, it appears that you did not conduct a stratified test, which every statistician would require following a stratified (by center) analysis. Please correct this, or specify that you did this already.

We now mentioned in the methods section (statistical analysis): ”Multivariable linear regression analysis was employed to check whether results were sensitive to controlling for baseline characteristics, including center of inclusion.” 

Results (in terms of betas and precision) were robust to such adjustment. Therefore the last line of the Results section says: 

“Adjustment, by means of multiple regression, for the small differences in baseline characteristics (augmentation, episiotomy, analgesics) or controlling center of inclusion did not result in any meaningful changes in the effect estimates or in more precision.” 

4. The DMC does not "abandon the trial prematurely", they determine if there is sufficient evidence to stop a trial early for efficacy, futility, and/or safety.

Adjusted

5. Throughout the paper you refer to the groups with the phrase "there is no difference". As you know, determination of a statistical difference requires more than just observation of numbers. It is more appropriate to discuss that the data appear to be comparable with respect to the given metric.

Adjusted throughout

6. The discussion section should (as noted above) indicate whether the assumptions in the sample size and power computations were realized in the study (e.g., effect size, LTF assumptions, etc.)

In results: “Correction for the small differences in baseline characteristics (augmentation, episiotomy, analgesics) did not result in any meaningful changes in the effect estimates or more precision.”

In discussion: “We reached our calculated sample size in both resuscitation arms and had little loss to follow-up. Baseline characteristics were well balanced and adjustment for small differences in the baseline gave similar results.”

Loss to follow up can only be applied for T3 results as women may already have left the hospital and this may also be viewed as missing data. None of the women were lost to follow up for the primary outcome. 

7. I have no idea how you would up with a 130:122 imbalance in treatment assignments given that you were using permuted blocks with a fixed size of 4. The maximum imbalance possible is 2 due to an unfilled block, or where there non-compensating incomplete blocks in multiple centers.

The envelopes were available at a central point at the labor ward in all locations. However these were taken to the operating theatre in case of a cesarian section. The operating theaters are not at the labor ward but in a different section of the hospital (at all participating locations) making it impossible and time consuming to get randomization envelope in case there was 500 mL and ongoing blood loss. The missing envelopes causing the imbalance as mentioned by this reviewer, were not opened but were disposed of during clean up instead of returned to their original central point.

8. You do not discuss the clinical centers, where they were, what their staffing was, how many patients they each recruited, and how study procedures were standardized.

Added, see lines 108-112 and 215-218

9. Fixed block sizes are frowned upon by statisticians--I presume the investigators were blinded to the block size to mitigate selection bias, and that should be stated if so.

Stated already in original manuscript:

 “Sequence was generated online (https://www.randomizer.org/) and the sealed opaque envelopes were created by an independent research nurse or medical student not involved in the randomization of the patient.”

Clarified with “Local investigators were blinded to block size and allocation.”

Reviewer #3: Dear Author,

I reported my comments on the manuscript titled “Restrictive versus liberal fluid resuscitation strategy, influence on blood loss and hemostatic parameters in mild obstetric hemorrhage: an open-label randomized controlled trial. (REFILL study)”.

I think that the topic is interesting. References list appears up to date and appropriate. The manuscript is clear and easy to understand also for non-specialized reader.

Without doubts, the study have many limits, some of which punctually explained in the manuscript. In my opinion a great limit is the selection of population, there are many confounding factor, for example the choose to recruit women both after vaginal delivery and cesarean section or the recruitment of women with risk factors for PPH. Probably the best way is recruit women without risk factors after a spontaneous vaginal delivery. Anyway, this limit is partially overcame by the fact that the sampling method is the same in both groups and by the decision to analyze data according to the intention-to-treat principle. Before the acceptance I suggest some minor revisions.

You (at line 143-146) explain that blood loss was measured by weighing the absorption towel after childbirth, but the first towel was disposed directly after childbirth and not measured as this also includes amniotic fluid. I suggest to explain how do you have solved the problem of amniotic fluid interference in case of cesarean section, in which the blood loss was measured through suction and weighing operative gauze.

In the operating theatre the nurse notes the amount of amniotic fluid suctioned during child birth. After child birth blood loss is counted from there onwards. Adjusted at line 147

In the methods, at line 164, I suggest to explain the actions performed in the active management of third stage of labor, for example the type and dosage and route of administration of uterotonic agent. 

See line 167-171 for adjustment

I suggest to check the flow diagram in the figure 1, because in my opinion could be some errors in the number of patient. Authors say that they have assessed for eligibility 5190 women. After that 1622 women signed informed consent and 3568 women were excluded. In the reasons of exclusion we have 1370 women that not met inclusion criteria and 1946 that declined to participate. In this way we have 252 lacking women (1370+1946=3316, 252 women less of 3568 women excluded). I think that this 252 women are not the women randomized, because the 252 women randomized were in the group of women that signed the consent. So I suggest to check the data, because if the 252 women randomized are added to the women that signed the consent we have not 1622 women but 1874 women. In this way the authors have to check also the data at line 339-340, because the rate 102/1874 is 5,4% in line with the litterature data.

Thank you for noticing, We reran the numbers and corrected those were needed. The 1622 is the correct number therefore 339-340 didn’t need adjustment. 

In the discussion at line 270, remove "low risk PPH", because the women in the study are not at low risk PPH. I think that you refer about the early and mild PPH (500 mL blood loss).

Adjusted

After these minor revisions, I suggest to accept this manuscript because in my personal opinion the paper is interesting for at least 3 reasons: the originality (first study on this topic), the study design (prospective, blinded, randomized controlled multicenter clinical trial) and for the results (which certainly deserve to be then confirmed on a larger scale).

6. PLOS authors have the option to publish the peer review history of their article (what does this mean?). If published, this will include your full peer review and any attached files.

Do you want your identity to be public for this peer review? For information about this choice, including consent withdrawal, please see our Privacy Policy.

Reviewer #1: No

Reviewer #2: No

Reviewer #3: No

---

## [Decision Letter · Decision Letter 1]

12 May 2021

PONE-D-21-04160R1

Restrictive versus liberal fluid resuscitation strategy, influence on blood loss and hemostatic parameters in mild obstetric hemorrhage: an open-label randomized controlled trial. (REFILL study)

PLOS ONE

Dear Dr. Schol,

Thank you for submitting your manuscript to PLOS ONE. After careful consideration, we feel that it has merit but does not fully meet PLOS ONE’s publication criteria as it currently stands. Therefore, we invite you to submit a revised version of the manuscript that addresses the points raised during the review process.

ACADEMIC EDITOR: The paper has undergone extensive revision according to the reviewers' comments. Some minor modifications should be done (see comments Reviewer 2) about the randomization procedure, some sentences should be added in the weaknesses of the study.

We look forward to receiving your revised manuscript.

Kind regards,

Guillaume Ducarme, MD, MSc, PhD

Academic Editor

PLOS ONE

Journal Requirements:

Reviewers' comments:

Reviewer's Responses to Questions

**Comments to the Author**

1. If the authors have adequately addressed your comments raised in a previous round of review and you feel that this manuscript is now acceptable for publication, you may indicate that here to bypass the “Comments to the Author” section, enter your conflict of interest statement in the “Confidential to Editor” section, and submit your "Accept" recommendation.

Reviewer #1: All comments have been addressed

Reviewer #2: (No Response)

Reviewer #3: All comments have been addressed

2. Is the manuscript technically sound, and do the data support the conclusions?

Reviewer #1: Yes

Reviewer #2: (No Response)

Reviewer #3: Yes

3. Has the statistical analysis been performed appropriately and rigorously? 

Reviewer #1: Yes

Reviewer #2: (No Response)

Reviewer #3: Yes

4. Have the authors made all data underlying the findings in their manuscript fully available?

Reviewer #1: Yes

Reviewer #2: (No Response)

Reviewer #3: Yes

5. Is the manuscript presented in an intelligible fashion and written in standard English?

Reviewer #1: Yes

Reviewer #2: (No Response)

Reviewer #3: Yes

6. Review Comments to the Author

Reviewer #1: (No Response)

Reviewer #2: Since the randomization procedure was compromised due to unopened envelopes, the explanation given in response to my comments should be added to the "Weaknesses of the study" section in the Conclusions. I would also add when stating the unbalanced sample size that "reasons for the imbalance are discussed in the conclusions", or discuss the issue earlier when you talk about the envelope procedure.

Reviewer #3: Congratulations for the manuscript! Interesting Topic, well conducted and written and understandable and of general interest paper.

7. PLOS authors have the option to publish the peer review history of their article (what does this mean?). If published, this will include your full peer review and any attached files.

Reviewer #1: No

Reviewer #2: No

Reviewer #3: **Yes: **Alessandro Svelato

---

## [Author Response · Author response to Decision Letter 1]

31 May 2021

Comments to the Author

1. If the authors have adequately addressed your comments raised in a previous round of review and you feel that this manuscript is now acceptable for publication, you may indicate that here to bypass the “Comments to the Author” section, enter your conflict of interest statement in the “Confidential to Editor” section, and submit your "Accept" recommendation.

Reviewer #1: All comments have been addressed

Reviewer #2: (No Response)

Reviewer #3: All comments have been addressed

2. Is the manuscript technically sound, and do the data support the conclusions?

Reviewer #1: Yes

Reviewer #2: (No Response)

Reviewer #3: Yes

3. Has the statistical analysis been performed appropriately and rigorously?

Reviewer #1: Yes

Reviewer #2: (No Response)

Reviewer #3: Yes

4. Have the authors made all data underlying the findings in their manuscript fully available?

Reviewer #1: Yes

Reviewer #2: (No Response)

Reviewer #3: Yes

5. Is the manuscript presented in an intelligible fashion and written in standard English?

Reviewer #1: Yes

Reviewer #2: (No Response)

Reviewer #3: Yes

6. Review Comments to the Author

Reviewer #1: (No Response)

Reviewer #2: Since the randomization procedure was compromised due to unopened envelopes, the explanation given in response to my comments should be added to the "Weaknesses of the study" section in the Conclusions. I would also add when stating the unbalanced sample size that "reasons for the imbalance are discussed in the conclusions", or discuss the issue earlier when you talk about the envelope procedure.

We added this to our strength and limitations section of our manuscript, and 347-354. 

Reviewer #3: Congratulations for the manuscript! Interesting Topic, well conducted and written and understandable and of general interest paper.

7. PLOS authors have the option to publish the peer review history of their article (what does this mean?). If published, this will include your full peer review and any attached files.

Do you want your identity to be public for this peer review? For information about this choice, including consent withdrawal, please see our Privacy Policy.

Reviewer #1: No

Reviewer #2: No

Reviewer #3: Yes: Alessandro Svelato

---

## [Decision Letter · Decision Letter 2]

14 Jun 2021

Restrictive versus liberal fluid resuscitation strategy, influence on blood loss and hemostatic parameters in mild obstetric hemorrhage: an open-label randomized controlled trial. (REFILL study)

PONE-D-21-04160R2

Dear Dr. Schol,

We’re pleased to inform you that your manuscript has been judged scientifically suitable for publication and will be formally accepted for publication once it meets all outstanding technical requirements.

Kind regards,

Guillaume Ducarme, MD, MSc, PhD

Academic Editor

PLOS ONE

Additional Editor Comments (optional):

Reviewers' comments:

Reviewer's Responses to Questions

**Comments to the Author**

1. If the authors have adequately addressed your comments raised in a previous round of review and you feel that this manuscript is now acceptable for publication, you may indicate that here to bypass the “Comments to the Author” section, enter your conflict of interest statement in the “Confidential to Editor” section, and submit your "Accept" recommendation.

Reviewer #3: All comments have been addressed

2. Is the manuscript technically sound, and do the data support the conclusions?

Reviewer #3: (No Response)

3. Has the statistical analysis been performed appropriately and rigorously? 

Reviewer #3: (No Response)

4. Have the authors made all data underlying the findings in their manuscript fully available?

Reviewer #3: (No Response)

5. Is the manuscript presented in an intelligible fashion and written in standard English?

Reviewer #3: (No Response)

6. Review Comments to the Author

Reviewer #3: (No Response)

7. PLOS authors have the option to publish the peer review history of their article (what does this mean?). If published, this will include your full peer review and any attached files.

Reviewer #3: **Yes: **Alessandro Svelato

---

## [Editor Report · Acceptance letter]

18 Jun 2021

PONE-D-21-04160R2 

Restrictive versus liberal fluid resuscitation strategy, influence on blood loss and hemostatic parameters in mild obstetric hemorrhage: an open-label randomized controlled trial. (REFILL study) 

Dear Dr. Schol:

I'm pleased to inform you that your manuscript has been deemed suitable for publication in PLOS ONE. Congratulations! Your manuscript is now with our production department. 

Kind regards, 

on behalf of

Dr. Guillaume Ducarme 

Academic Editor

PLOS ONE